# Cross-protocol assessment of induction and durability of VISP/R in HIV preventive vaccine trial participants

Nicole Espy[1], Xue Han[1], Shannon Grant[1], Esther Kwara[2¤],
Bharathi Lakshminarayanan[1], Michael Stirewalt[1], Kelly E. Seaton[3], Georgia
D. Tomaras[3], Erin Goecker[4], Julie McElrath[1,4], Jessica Andriesen[1], Yunda Huang[1,5],
Stephen R. Walsh[2], John Hural[1] *

1 Vaccine and Infectious Diseases Division, Fred Hutchinson Cancer Center, Seattle, Washington, United States of America, 2 Division of Infectious Diseases, Brigham & Women's Hospital, Harvard Medical School, Boston, Massachusetts, United States of America, 3 Department of Surgery, Duke University, Durham, North Carolina, United States of America, 4 Department of Laboratory Medicine and Pathology, University of Washington, Seattle, Washington, United States of America, 5 Department of Global Health, University of Washington, Seattle, Washington, United States of America

¤ Current address: Department of Pediatrics, Northwestern Medical School, Chicago, Illinois, United States of America

* jhural@fredhutch.org

**Data Availability Statement:** Data used from this analysis is available on the ATLAS data portal (https://atlas.scharp.org/cpas/project/HVTN%

## Abstract

Candidate HIV vaccines are designed to induce antibodies to various components of the HIV virus. An unintended result of these antibodies is that they may also be detected by commercial HIV diagnostic kits designed to detect an immune response to HIV acquisition. This phenomenon is known as Vaccine-Induced Seropositivity/Reactivity (VISP/R). In order to identify the vaccine characteristics associated with VISP/R, we collated the VISP/R results from 8,155 participants from 75 phase 1/2 studies and estimated the odds of VISP/R by multivariable logistic regression and 10-year estimated probability of persistence in relation to vaccine platform, HIV *gag* and *envelope* (*env*) gene inserts, and protein boost. Recipients of viral vectors, protein boosts, and combinations of DNA and viral-vectored vaccines had higher odds of VISP/R compared to those who received DNA-only vaccines (odds ratio, OR = 10.7, 9.1, 6.8, respectively, p<0.001). Recipients of gp140+ *env* gene insert (OR = 7.079, p<0.001) or gp120 *env* (OR = 1.508, p<0.001) had higher odds of VISP/R compared to those participants who received no *env*. Recipients of gp140 protein had higher odds of VISP/R than those that did not receive protein (OR = 25.155, p<0.001), and recipients of gp120 protein, had lower odds of VISP/R than those that did not receive protein (OR = 0.192, p<0.001). VISP/R persisted at 10 years in more recipients of *env* gene insert or protein compared to those who did not (64% vs 2%). The inclusion of *gag* gene in a vaccine regimen had modest effects on these odds and was confounded by other covariates. Participants receiving gp140+ gene insert or protein were most often reactive across all serologic HIV tests. Conclusions from this association analysis will provide insight into the possible impact of vaccine design on the HIV diagnostic landscape and vaccinated populations.

20Public%20Data/begin.view?) and can be provided upon request.

**Funding:** This work was supported by the National Institutes of Health (UM1 AI068614 to GDT; UM1 AI068618 to JM; UM1 AI068635 to XH, SG, YH; UM1 AI069412 to SRW; UL1 RR025758 to SRW; P30 AI064518 to GDT). MA and PD The funders had no role in study design, data collection and analysis, decision to publish, or preparation of the manuscript.

**Competing interests:** I have read the journal's policy and the authors of this manuscript have the following competing interests: SRW is an Academic Editor at PLoS One; the authors declare no other competing interests.

## Introduction

Nearly 40 years after the discovery of HIV, the virus continues to disproportionately burden vulnerable countries and groups. The World Health Organization (WHO) estimated that 1.5 million new infections occurred in 2020 [1], signifying that a safe and effective HIV vaccine remains a priority in curbing this pandemic.

More than three decades of HIV vaccine research has led to the clinical evaluation of many candidate regimens, though to date none have resulted in an efficacious product fit for licensure. These vaccines vary in platform (e.g., DNA or viral vector), gene insert (e.g., HIV *gag* or *envelope* [*env*]), protein boost (e.g., Env glycoprotein 120 [gp120] or gp140), and adjuvant (e.g., alum or MF59). Antibodies elicited to HIV antigens among participants receiving these candidate vaccine regimens may linger past the observation period of the clinical trial, which is typically only 6–12 months after the last vaccination [2]. Importantly, these antibodies may be detected by commercial HIV serologic diagnostic kits which aim to detect HIV-elicited antibodies and contain similar antigens to the vaccine regimens evaluated thus far: HIV p24, encoded by *gag*, and Env glycoprotein. Thus, these antibodies can confound the interpretation of serologic test results and subsequent diagnosis of an actual HIV acquisition, a phenomenon known as vaccine induced seropositivity/reactivity (VISP/R). Diagnostic tests that directly detect components of the HIV virus, such as nucleic acid amplification tests, are needed to differentiate a true HIV infection from VISP/R. VISP/R affects the trial participant's ability to receive a timely and accurate diagnosis of their HIV status from healthcare providers unfamiliar with the diagnostic complications that arise from vaccine-induced antibodies. It may also result in detrimental social impacts to trial participants, such as complications obtaining health insurance, donating blood or organs, or serving in the military [3]. If an efficacious vaccine for HIV prevention also induces VISP/R, the successful deployment of this vaccine could be compromised unless diagnostic tests agnostic to VISP/R are concurrently developed, commercialized, and adopted globally.

A cross-sectional analysis of the prevalence of VISP/R in HIV vaccine recipients in 25 HIV Vaccine Trial Network (HVTN) studies was performed in 2010 [4]. The general findings of this analysis were that VISP/R varied by vaccine product, and that inclusion of *env* or *gag* inserts increased the rate of VISP/R in study participants. However, this analysis did not measure the correlation of specific vaccine characteristics with the increased the rate of VISP/R, nor the duration for which participants would need HIV testing that differentiates true HIV infection from VISP/R. Since 1999, the HVTN has evaluated the safety and efficacy of new vaccine regimens and employed up-to-date HIV diagnostic testing platforms. From 2011 to 2020, the HVTN conducted HVTN 910, a longitudinal observational study which measured the duration of VISP/R and its social impacts in participants who received a vaccine in preventative HIV vaccine trials. Therefore, further evaluation of which vaccine characteristics are associated with the occurrence of VISP/R and its duration in participants is now possible and warranted.

In this analysis, we assessed the occurrence of VISP/R across 75 HIV vaccine studies to identify vaccine characteristics associated with this phenomenon. We then estimated the duration of VISP/R in participants enrolled in HVTN 910, grouped by characteristics of the vaccine regimen they received in their parent protocol. We also assessed the relationship between vaccine-induced binding antibody titers and the occurrence of VISP/R. These analyses will help identify components of an experimental HIV vaccine regimen that may induce VISP/R in study participants and inform the impact an efficacious preventative HIV vaccine would have on the HIV diagnostic landscape.

## Methods

### Ethics statement

Participants received HIV testing during their participation in these trials in accordance with Centers for Disease Control and Prevention (CDC) and local guidelines. On-site testing results could be blinded when necessary. An end of study visit was usually performed 6–12 months after the last vaccination per study guidelines, or retrospectively at the last visit before loss-to-follow-up, to identify participants who had acquired VISP/R. HVTN provided participants with VISP/R post-study HIV testing until VISP/R was no longer detected.

All trials were approved by the institutional review board/ethics committees of each clinical research site (CRS) institution prior to participant screening and enrollment. All participants provided written informed consent for both the parent protocol and HVTN 910.

### Study setting

We performed a longitudinal analysis of data collected from 75 HIV placebo-controlled, multi-center, double-blind, randomized vaccine trials from 1990 to 2020 in which participants were given a study product intended for preventing HIV acquisition: five phase 1 AIDS Vaccine Evaluation Group (AVEG) studies, 64 phase 1-2a HVTN studies, and 6 phase 2b HVTN studies (Table 1). Participants in these vaccine trials received preventative HIV vaccine regimens varying by DNA or viral vectors platform (VV; derived from alphavirus, adenovirus, poxvirus, or vesicular stomatitis virus), recombinant proteins, or a combination of the above. HIV-derived gene inserts primarily included *env*, *gag*, or both. Envelope gene insert and protein lengths were grouped into two categories: gp120 (sometimes linked to the gp41 transmembrane anchor peptide, gp41TM) and gp140+ (inclusive of gp120 with subunits of the gp41 ectodomain, gp140, gp145, gp150, gp160).

### Cohort

From 2011 to 2020, a total of 21,578 participants from the Americas, Africa, and Southeast Asia were enrolled in the 75 HIV vaccine trials included in this analysis. Eligibility criteria for participants included having a low likelihood of HIV acquisition for most phase 1 and 2a studies and a high likelihood of HIV acquisition for phase 2b studies. Participants were healthy, HIV-seronegative adults between 18 and 60 years old at the time of enrollment. VISP/R testing was performed at or near the end of the protocol. To assess the vaccine characteristics associated with the occurrence of VISP/R, participants were excluded from this analysis if they received a placebo/control product (n = 8,397), did not undergo a VISP/R assessment in their parent protocol (n = 4,833), acquired HIV by end of study (n = 39), or received study product but terminated early or met other exclusion criteria (n = 154). Thus, this analysis was performed on a total of 8,155 participants (Fig 1).

VISP/R was detected in 4,290 participants at the end of their parent protocols. These individuals were offered prospective, unscheduled HIV and VISP/R testing and were eligible to enroll in HVTN 910, a prospective observational trial designed to monitor of the persistence of VISP/R. Participants from previously completed AVEG protocols were also eligible to enroll in HVTN 910. To join HVTN 910, participants needed to have access to an active HVTN CRS, be willing to receive pre- and post-test HIV counseling and HIV results, demonstrate understanding of the study, and have no conditions that would be a contraindication to protocol adherence or the ability to give informed consent (in the judgement of the CRS investigator).

Of the eligible AVEG and HVTN participants, 1,146 were enrolled in HVTN 910. A total of 21 participants had no end of study VISP/R visit and their first record is within HVTN 910.

**Table 1. Protocols by HIV vaccine regimen characteristic.**

| Vaccine Platform | Platform Details | env insert | Env Protein | Protocol numbers* |
|---|---|---|---|---|
| DNA | DNA | No Env | No Protein | HVTN 045 (NCT00043511), HVTN 060 (NCT00111605), HVTN 063 (NCT00115960), HVTN 119 (NCT03181789) |
| | | gp120 +gp41TM | No Protein | HVTN 044 (NCT00069030), HVTN 052 (NCT00071851) |
| | | gp140 | No Protein | HVTN 092 (NCT01783977) |
| | | | gp120 | HVTN 105 (NCT02207920), HVTN 108 (NCT02915016), HVTN 111 (NCT02997969) |
| | | | gp140 | HVTN 049 (NCT00073216) |
| | | gp150 | No Protein | HVTN 070 (NCT00528489), HVTN 080 (NCT00991354), HVTN 098 (NCT02431767) |
| DNA.VV | DNA/Ad35 | gp140 | No Protein | *HVTN 072 (NCT00472719), HVTN 077 (NCT00801697)* |
| | DNA/Ad5 | gp120TM/ gp140 | No Protein | HVTN 068 (NCT00270218), HVTN 069 (NCT00384787), HVTN 082 (NCT01054872), HVTN 204 (NCT00125970) |
| | | gp140 | No Protein | HVTN 057 (NCT00091416), *HVTN 072 (NCT00472719)*, HVTN 076 (NCT00955006), *HVTN 077 (NCT00801697)* |
| | | gp140/gp145 | No Protein | HVTN 505 (NCT00865566) |
| | DNA/MVA | No Env | No Protein | HVTN 065 (NCT00301184) |
| | | gp150 | No Protein | HVTN 073 (NCT00574600), HVTN 086 (NCT01418235) |
| | | gp150/gp160 | No Protein | HVTN 094 (NCT01571960), HVTN 106 (NCT02296541), HVTN 205 (NCT00820846) |
| | | gp160 | No Protein | HVTN 114 (NCT02852005) |
| | DNA/MVA/ protein | gp150 | gp140 | HVTN 073 (NCT00574600), HVTN 086 (NCT01418235) |
| | | gp160 | gp120 | HVTN 114 (NCT02852005) |
| | DNA/NYVAC | gp140 | No Protein | HVTN 092 (NCT01783977) |
| | | | gp120 | HVTN 096 (NCT01799954) |
| | DNA/VSV | gp140/gp160 | No Protein | *HVTN 112 (NCT02654080)* |
| | | gp160 | No Protein | HVTN 087 (NCT01578889) |
| Protein | Protein | No Env | gp120 | *AVEG 005B (NCT00000632), HVTN 041 (NCT00027365), HVTN 108 (NCT02915016), HVTN 110 (NCT02771730)* |
| | | | gp140 | *HVTN 049 (NCT00073216), HVTN 073 (NCT00574600), HVTN 088 (NCT01376726)* |
| | | | gp145 | *HVTN 122 (NCT03382418)* |
| | | | gp160 | *AVEG 003 (NCT00000745), AVEG 004 (NCT00000968)* |

(*Continued*)

**Table 1.** (Continued)

| Vaccine Platform | Platform Details | env insert | Env Protein | Protocol numbers* |
|---|---|---|---|---|
| VV | Ad26/Ad35 | gp140 | No Protein | *HVTN 091 (NCT01215149)* |
| | Ad26/protein | gp140 | gp140 | HVTN 117 (NCT02788045) |
| | | | | HVTN 118 (NCT02935686) |
| | Ad35 | gp140 | No Protein | *HVTN 083 (NCT01095224)* |
| | Ad4/protein | No Env | gp120 | *HVTN 110 (NCT02771730)* |
| | | gp150 | gp120 | *HVTN 110 (NCT02771730)* |
| | Ad5 | No Env | No Protein | HVTN 050 (NCT00849732), HVTN 071 (NCT00486408), HVTN 084 (NCT01159990), HVTN 502 (NCT00095576), HVTN 503 (NCT00413725), HVTN 504 (n/a) |
| | | gp140 | No Protein | HVTN 054 (NCT00119873), HVTN 068 (NCT00270218), *HVTN 083 (NCT01095224)*, HVTN 084 (NCT01159990), HVTN 085 (NCT01479296) |
| | Ad5/Ad35 | gp140 | No Protein | *HVTN 072 (NCT00472719), HVTN 077 (NCT00801697), HVTN 083 (NCT01095224)* |
| | Ad5/NYVAC | gp120/gp140 | No Protein | HVTN 078 (NCT00961883) |
| | Alphavirus | No Env | No Protein | HVTN 040 (NCT00063778), HVTN 059 (NCT00097838) |
| | ALVAC | gp120 | No Protein | HVTN 026 (NCT00011037), HVTN 039 (NCT00027261), HVTN 203 (NCT00007332) |
| | | | gp120 | HVTN 026 (NCT00011037), HVTN 097 (NCT02109354), HVTN 100 (NCT02404311), HVTN 107 (NCT03284710), HVTN 203 (NCT00007332) |
| | | gp120 +gp41TM | No Protein | HVTN 042 (NCT00076063) |
| | | | gp120 | HVTN 120 (NCT03122223), HVTN 702 (NCT02968849) |
| | | gp140 | gp160 | HVTN 034 (n/a) |
| | | | p24 | HVTN 032 (n/a) |
| | FPV | gp140 | No Protein | HVTN 055 (NCT00083603) |
| | MVA | gp140 | No Protein | HVTN 055 (NCT00083603) |
| | | gp150 | No Protein | HVTN 065 (NCT00301184), HVTN 205 (NCT00820846) |
| | | | gp140 | HVTN 086 (NCT01418235) |
| | | gp160 | No Protein | HVTN 114 (NCT02852005) |
| | | | gp120 | HVTN 114 (NCT02852005) |
| | MVA/FPV | gp140 | No Protein | HVTN 055 (NCT00083603) |
| | NYVAC/ protein | gp140 | gp120 | HVTN 096 (NCT01799954) |
| | Vaccinia/ Protein | gp160 | gp160 | *AVEG 002 (NCT00000683), AVEG 002B (NCT00000631)* |
| | VSV | No Env | No Protein | HVTN 090 (NCT01438606) |

*Italicized protocols contained *gag* genes or epitopes.

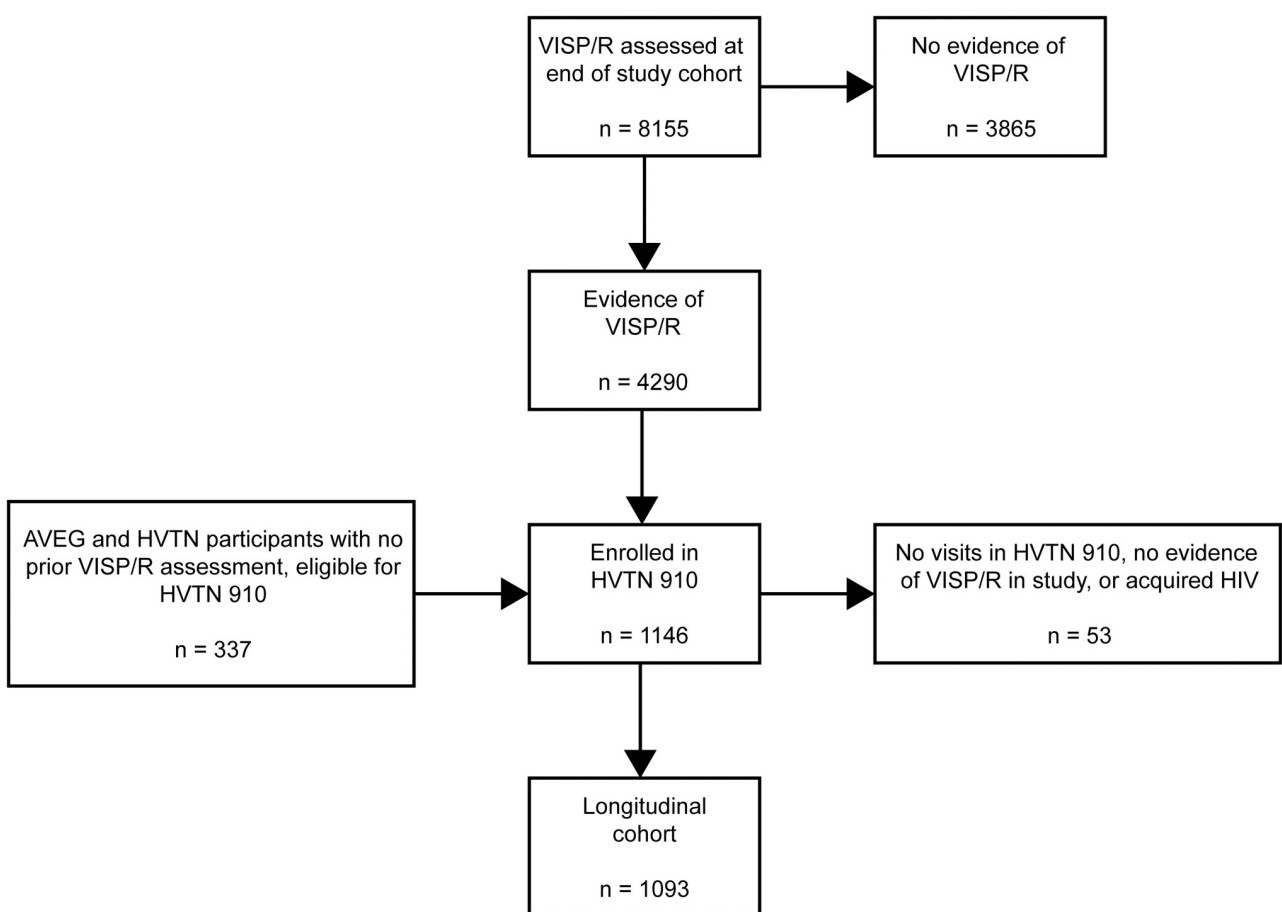

**Fig 1. Study flow diagram displaying participants analyzed for VISP/R in experimental HIV vaccine trials and HVTN 910.** Participants were enrolled in HVTN vaccine trials and evaluated for VISP/R at the end of the study. Those that received vaccine product were included in the analysis to identify vaccine characteristics associated with the occurrence of VISP/R. Participants with VISP/R were eligible to enroll in the long-term observational trial HVTN 910 to observe their VISP/R status until it resolved. AVEG participants and those in HVTN 032, 034, 050, 057, 077, 502, 503, in which no previous VISP/R assessment was performed, were also eligible to enroll in HVTN 910. Participants that had detectable VISP/R in study comprised the longitudinal cohort analyzed to estimate the duration of VISP/R.

Participants who had no detectable VISP/R during HVTN 910, acquired HIV while enrolled, or had no VISP/R visits after enrollment were excluded from the analysis of VISP/R persistence. Participants were offered follow-up HIV testing (recommended approximately every six months) until the resolution of VISP/R. HVTN 910 was closed on 01 October 2020 and all participants with persistent VISP/R are currently provided post-study testing though standard HVTN processes.

## Laboratory testing

VISP/R testing was performed according to a preapproved HIV diagnostic algorithm at three Division of AIDS (DAIDS)-approved Good Clinical Laboratory Practice (GCLP)-compliant laboratories: the University of Washington Retrovirology Laboratory in Seattle, Washington, which performed testing of specimens from Europe, North America, and the Western Pacific (WP); the National Institute of Communicable Diseases in Johannesburg, South Africa, which performed testing of specimens from Africa; and the Asociación Civil Impacta Clinical Trials Unit (CTU) HIV Diagnostics Lab in Lima, Peru, which performed testing of specimens from

South America. The Viral and Rickettsial Disease Laboratory at the California Department of Health Services (CL-Richmond) also performed diagnostic testing of specimens from North America prior to 2006.

VISP/R was defined as a participant specimen resulting in a reactive result from a serological HIV laboratory screening or rapid test (i.e., antigen/antibody (Ag/Ab) combination or antibody enzyme immunoassay/chemiluminescent microparticle immunoassay (EIA/CMIA) test) and undetectable HIV-1 RNA by PCR. The VISP/R testing algorithm includes the performance of three or more independent U.S. Food and Drug Administration (FDA)-approved/ Conformité Européene (CE)-marked HIV laboratory screening or rapid tests, followed by an HIV-1 RNA PCR if any positive or reactive laboratory screening or rapid results are obtained. To differentiate a possible HIV acquisition from VISP/R, participant specimens that elicited reactive laboratory screening or rapid and HIV-1 RNA negative results could also be tested by Western Blot or HIV-1/2 differentiation tests according to criteria by the CDC and the manufacturer's package insert. The laboratory screening or rapid tests were selected by how commonly they were used in the participant's regional diagnostic laboratories. The list of diagnostic tests for each laboratory is presented in Table 2. Testing was performed according to the manufacturer's package insert.

## Statistical analysis

Participant demographics are summarized by each protocol and pooled. Geographic regions were consistent with WHO definitions, with Europe, the Americas, and WP grouped into one region since specimens were tested on comparable diagnostic platforms. Race categories were defined in accordance with United States Census race categories.

For each parent protocol, all enrolled vaccine recipients (i.e., those who received at least one dose of vaccine) were included in the analysis. VISP/R rates are summarized by demographics and vaccine characteristics, with 95% confidence intervals (CIs) computed using the Wilson score method [5]. Logistic regression models were applied to study the association between VISP/R response rates and vaccine characteristics, including vaccine product, gene insert, and protein. A multivariable logistic regression model was developed to examine the association between VISP/R status and vaccine characteristics, adjusted for age, region, trial phase and sex assigned at birth.

VISP/R duration analysis included all participants enrolled in HVTN 910 who had detectable VISP/R at the end of the parent protocol. Per-protocol VISP/R resolution in HVTN 910 was defined as having three consecutive seronegative tests over at least one year, followed by no subsequent positive tests. The per-protocol time to resolution was measured from the date of last vaccination in the parent protocol. For those whose VISP/R resolved, the event time was estimated as the midpoint between the date of last detection of VISP/R and first of three consecutive seronegative tests. Participants with only one or two seronegative tests were censored at the midpoint of the last positive and first negative test date. For participants who tested seropositive at their most recent HVTN 910 visit, time to VISP/R resolution was censored at the date of their last visit. Due to incomplete participant follow-up, we also considered a modified VISP/R resolution definition as having at least one seronegative test followed by no subsequent seropositive tests. The modified time to resolution was also measured from last vaccination, with event time defined as the midpoint between the date of the last detection of VISP/R and the first seronegative test.

Kaplan-Meier estimates of VISP/R persistence were calculated within study product categories. Median time to resolution and resolution rates at 2, 5, and 10 years were summarized within study product categories.

**Table 2. VISP/R rate per diagnostic assay.**

| Laboratory Screening Tests | Years of Use | HIV diagnostics lab | gp120 env gp120 protein + gag | gp120 env No Protein + gag | gp120 env No Protein No gag | gp140+ gp120 protein + gag | gp140+ gp120 protein No gag | gp140+ gp140 + protein + gag | gp140+ No protein + gag | gp140+ No protein No gag | No Envelope gp120 protein + gag | No Envelope gp120 protein No gag | No Envelope gp140 + protein No gag | No Envelope No Protein + gag |
|---|---|---|---|---|---|---|---|---|---|---|---|---|---|---|
| Abbott Architect HIV Ag/Ab Combo ** | 2011-present | UWVSL, NICD | 1% | | | 15% | 25%* | 99% | 79% | 91% | 0%* | 8% | 93% | 4% |
| Abbott Axsym HIV Ag/Ab Combo** | 2011–2014 | NICD | | | | | | 93% | 53%* | 98%* | | | | 0%* |
| Abbott Murex HIV-1.2.O | 2008–2011 | NICD | | | | | | | 77%* | | | | | 70% |
| Abbott HIV AB HIV1/2 (rDNA) | 2006–2011 | UWVSL | 61% | 54% | 3%* | | | 100%* | 85% | 66% | | 0% | 100%* | 60% |
| Abbott Prism Anti HIV-1/2, PSBC* | 2013–2021 | UWVSL | 0% | | | 16% | 0%* | 50%* | 63% | 84% | 0%* | 0%* | 86%* | 1% |
| bioMerieux Vironostika HIV Ag/Ab HIV 1/2 | 2011–2013 | NICD | | | | | | 91% | 56%* | 98% | | | | 0%* |
| bioMerieux Vironostika HIV Uni-Form II + O | 2008–2011 | NICD | | | | | | | 64%* | | | | | 70% |
| bioMerieux Vironostika HIV-1 | 2006–2007 | UWVSL | 62% | 56% | 8%* | | | 100% | 87% | 0%* | | | 100%* | 17% |
| BioRad GS HIV Combo Ag/Ab EIA | 2013-present | UWVSL, Impacta | 2% | | | 24% | 25%* | 98% | 76% | 82%* | 0%* | 13%* | 97% | 6% |
| BioRad GenScreen Ultra HIV Ag-Ab HIV 1/2 | 2011-present | NICD | 2% | | | 1% | | 97% | 54%* | 98% | | 0%* | | 0%* |
| BioRad Genetic Systems HIV 1/2 Plus O EIA | 2006–2013 | UWVSL | | 43%* | | 0%* | | 100%* | 79% | 66% | | | 96% | 47% |
| BioRad GenScreen HIV 1/2 | 2008–2011 | NICD | | | | | | | 79%* | | | | | 76% |
| BioRad Genetic Systems rLAV | 2007–2011 | UWVSL | | | | | | | 74% | 64% | | | | 52% |
| **Rapid Tests** | | | | | | | | | | | | | | |
| Alere Determine HIV-1/2 Ag/Ab Combo | 2016-present | UWVSL, NICD, Impacta | 2% | | | 7% | 100%* | 99% | 74% | 82%* | | 7%* | 95%* | 6%* |
| Abbott Determine HIV Early Detect | 2022-present | NICD | | | | | | 100%* | | | | | | |
| Alere HIV Combo | 2019–2021 | NICD | 1% | | | 0%* | | 100%* | | | | 0%* | | |
| Biorad Multispot HIV-1/HIV-2 Rapid Test | 2009–2017 | UWVSL, NICD | 1% | | | 32% | 0%* | 93% | 79% | 92% | 0%* | 33%* | 92%* | 27% |
| SD Bioline HIV-1/2 3.0 | 2020–2020 | NICD | 0% | | | | | | | | | | | |
| **Western Blot/ Differentiation tests** | | | | | | | | | | | | | | |
| BioRad Geenius HIV 1/2 Confirmatory Assay | 2018-Present | UWVSL | 100%* | | | 81%* | | 100% | 100%* | | | 100%* | 100%* | 75%* |
| BioRad Genetic Systems HIV-1 | 2006–2016 | UWVSL, NICD | | 86%* | | 100%* | | 100% | 93% | 82% | | | 100% | 64% |
| Richmond In-House Western Blot | 2000–2006 | CL-Richmond | 95% | 97% | | | | 100%* | 100%* | | | | | 100%* |

*VISP/R rate calculated from <50 participants. Color scale applied; red equals 0%, yellow 50%, and green 100%.

Assessments of the association between vaccine-induced immunogenicity, as measured by total IgG, to antibody reactivity to HIV diagnostic tests was performed using available binding antibody multiplex assay (BAMA) data previously collected from vaccine recipients in protocols HVTN 098, 100, 106, 107, 108, 111, 112, 114, 117, 118, 122, 205, and 505 [6–10]. Total IgG binding antibody responses were measured at two to four weeks following the last scheduled vaccination using matched antigen lots of gp120 Group B (Con 6 gp120/B), consensus gp140 (Con S gp140 CFI), variable loop 1 and 2 (V1/V2) (gp70_B.CaseA_V1_2), and gp41 antigens. Statistical analyses were performed using SAS (version 9.4, SAS Institute, Cary, NC) and R statistical software (version 3.5.3 R Foundation for Statistical Computing, Vienna, Austria).

## Results

### Risk of VISP/R in HIV preventative vaccine trial participants

Participant demographics including age, sex at birth, and race reflected the target populations enrolled for each geographic region (Table 3). Approximately 75% of these participants (6,205) were enrolled in the Americas, Europe, or the WP, of which less than 3% were enrolled in Europe and the WP. Of these 6,205 participants, 2,735 (44%) were under 30 years old, 3,863 (62%) were assigned male at birth, 3,873 (63%) identified as White, and 5,423 (87%) identified as non-Hispanic. Of the 1,950 participants enrolled in sub-Saharan Africa, 1,589 (81%) were under 30 years old, 991 (51%) were assigned male at birth, 1,869 (96%) identified as Black, and all identified as either non-Hispanic (1362, 70%) or were of unknown ethnicity (588, 30%).

**Table 3. Demographics of participants with VISP/R status assessed at the end of their parent protocol.**

| | Combined | Sub-Saharan Africa | Americas/Europe/ WP |
|---|---|---|---|
| | **N = 8155** | **N = 1950** | **N = 6205** |
| **Age** | | | |
| 30 or above | 38% (3096) | 19% (361) | 44% (2735) |
| Less than 30 | 62% (5059) | 81% (1589) | 56% (3470) |
| **Sex assigned at birth** | | | |
| Female | 40% (3301) | 49% (595) | 38% (2342) |
| Male | 60% (4854) | 51% (991) | 62% (3863) |
| **Race/Ethnicity** | | | |
| American Indian or Alaska Native | 0% (36) | 0% (0) | 1% (36) |
| Asian | 2% (183) | 0% (2) | 3% (181) |
| Black or African American | 38% (3101) | 96% (1869) | 20% (1232) |
| Hispanic or Latino | 2% (157) | 0% (0) | 3% (157) |
| Multiracial | 6% (476) | 0% (0) | 8% (476) |
| Native Hawaiian or Other Pacific Islander | 0% (14) | 0% (0) | 0% (14) |
| Unknown | 4% (314) | 4% (78) | 4% (236) |
| White | 48% (3874) | 0% (1) | 62% (3873) |
| **Ethnicity** | | | |
| Hispanic | 9% (767) | 0% (0) | 12% (767) |
| Non-Hispanic | 83% (6785) | 70% (1362) | 87% (5423) |
| Unknown | 7% (603) | 30% (588) | 0% (15) |
| **Study Phase** | | | |
| Phase 1-2a | 64% (5208) | 58% (1123) | 66% (4085) |
| Phase 2b | 36% (2947) | 42% (827) | 34% (2120) |

Over half of all participants across geographic regions were enrolled in phase 1-2a studies (58% in sub-Saharan Africa, 66% in Americas/Europe/WP).

Study phase (phase 2b vs. phase 1/2a), region (Americas/Europe/WP vs. SSA), vaccine platform (non-DNA vs. DNA), and the presence of a gp140 gene or protein were associated with increased odds of VISP/R in multivariate analyses (Table 4). Participants in the Americas, Europe, and WP had increased odds of VISP/R (OR = 2.559, p<0.001) compared to those from sub-Saharan Africa. Participants enrolled in phase 2b studies had increased odds of VISP/R (OR = 2.028, p = <0.001) compared to those in phase 1-2a. VV only, VV in combination with DNA, and protein-only regimens had increased odds of inducing VISP/R compared to DNA alone. The inclusion of a gp120 or gp140 gene insert in vaccine regimens increased the odds of VISP/R compared to participants who did not receive an *env* gene insert (OR = 1.508 and 7.079 respectively, p<0.001). The inclusion of gp140+ protein (OR = 25.155, p<0.001) in vaccine regimens, whether alone or in combination with DNA or viral vectors, increased the odds of VISP/R compared to participants who did not receive gp140+ protein. In contrast, the inclusion of a gp120 protein boost reduced the odds of VISP/R in participants (OR = 0.192, p<0.001). The inclusion of a *gag* gene insert decreased the odds of VISP/R (OR = 0.752, p = 0.017) despite a higher rate of VISP/R in participants with *gag* (60.4% VISP/R in those with *gag* versus 51.8% in those without *gag*). This is likely due to confounding by *env*

**Table 4. Multivariate analysis of VISP/R rate.**

| Category | VISP/R rate | 95% CI | Multivariate Odds Ratio (95% CI) | P value |
|---|---|---|---|---|
| Age | | | | |
| 30 or above | 55.1% (1706/3096) | (53.3%, 56.9%) | Ref | 0.026 |
| Less than 30 | 51.1% (2584/5059) | (49.7%, 52.4%) | 1.142 (1.016, 1.284) | |
| Sex assigned at birth | | | | |
| Female | 47.0% (1550/3301) | (45.3%, 48.7%) | Ref | 0.646 |
| Male | 56.4% (2740/4854) | (55.0%, 57.8%) | 1.028 (0.914, 1.156) | |
| Study Phase | | | | |
| Phase 1-2a | 48.0% (2502/5208) | (46.7%, 49.4%) | Ref | <0.001 |
| Phase 2b | 60.7% (1788/2947) | (58.9%, 62.4%) | 2.028 (1.747, 2.355) | |
| Region | | | | |
| Sub-Saharan Africa | 26.8% (522/1950) | (24.9%, 28.8%) | Ref | <0.001 |
| Americas/Europe/WP | 60.7% (3768/6205) | (59.5%, 61.9%) | 2.559 (2.201, 2.975) | |
| Vaccine Platform | | | | |
| DNA | 14.6% (177/1213) | (12.7%, 16.7%) | Ref | <0.001 |
| DNA.VV | 77.2% (1874/2426) | (75.5%, 78.9%) | 6.84 (5.443, 8.596) | |
| Protein | 47.7% (103/216) | (41.1%, 54.3%) | 9.126 (4.474, 18.614) | |
| VV | 49.7% (2136/4300) | (48.2%, 51.2%) | 10.693 (8.456, 13.521) | |
| Gag | | | | |
| No *gag* | 51.8% (3853/7432) | (50.7%, 53.0%) | Ref | 0.017 |
| *gag* | 60.4% (437/723) | (56.8%, 63.9%) | 0.752 (0.595, 0.95) | |
| Env | | | | |
| No Env | 40.8% (980/2401) | (38.9%, 42.8%) | Ref | <0.001 |
| gp120 env | 21.3% (302/1415) | (19.3%, 23.5%) | 1.508 (1.205, 1.887) | |
| gp140+ env | 69.3% (3008/4339) | (67.9%, 70.7%) | 7.079 (5.773, 8.68) | |
| Protein | | | | |
| No protein | 60.3% (3563/5906) | (59.1%, 61.6%) | Ref | <0.001 |
| gp120 protein | 11.6% (197/1702) | (10.1%, 13.2%) | 0.192 (0.155, 0.238) | |
| gp140+ protein | 96.9% (530/547) | (95.1%, 98.0%) | 25.155 (13.981, 45.259) | |

and glycoprotein covariates since analysis of the inclusion of *gag*, adjusting for demographics and vaccine platform, demonstrated increased odds of VISP/R (OR = 1.634, p<0.001) (S1 Table). These observations confirm that regimens including gp140+ *env* gene inserts or protein boosts are more likely to induce specific seroreactive antibodies than regimens without these components.

## Estimating the risk of VISP/R by binding antibody titers to Env

To further characterize the relationship between vaccine Env antigens and seroreactive antibodies, we compared the occurrence of VISP/R for each vaccine platform or gene insert/protein combination to binding antibody responses. We chose antigens that were analyzed by BAMA for each study listed in the Methods: Con 6 gp120/B, Con S gp140, CaseA V1/V2, and gp41. Participants with VISP/R had higher binding antibody magnitudes than participants with no VISP/R for all antigens tested, regardless of vaccine platform or envelope gene insert/protein combination (Fig 2A). Multivariate logistic regression of participant age and sex assigned at birth, inclusion of *gag*, vaccine type, and binding antibody response magnitude indicate that responses to Con 6 gp120/B and CaseA V1/V2 result in a significant increase in the odds of VISP/R (S2 Table). There was no significant association between VISP/R and a response to gp140 antigen, likely due to insufficient gp140 antigen BAMA data from participants without VISP/R. A regression model to indicate if antibody response to gp41 affected the odds of VISP/R, adjusting for envelope/glycoprotein combinations, was unstable, likely due to lack of data for all combinations.

## Duration of VISP/R post-vaccination

Participants who had VISP/R detected in their parent protocol or were previously enrolled in an AVEG study were offered enrollment into the long-term follow-up study: HVTN 910. In total, 1,093 participants completed at least one visit in which VISP/R was identified, enrolled in HVTN 910, and were not later diagnosed as having acquired HIV (Fig 1 and S1 Fig). Demographics represent regional participant characteristics (S3 Table). Participants from all regions were enrolled in HVTN 910, with comparable years of follow-up since last vaccination, follow-up time from enrollment in HVTN 910, and median number of VISP/R assessments (Table 5). Of these participants, 1% from the Americas, Europe, and the WP and 4% from sub-Saharan Africa terminated from the study due to acquiring HIV. More participants from the Americas, Europe, and the WP terminated from the study due to study closure (75%) rather than resolution of VISP/R (10%). More participants from sub-Saharan Africa terminated from the study due to resolution of VISP/R (45%) than from study closure (40%). Due to incomplete follow-up of participants, we measured the frequency at which participants met a modified definition of VISP/R resolution as described above. For all groups, more participants had met a modified VISP/R resolution than met the original per-protocol definition of VISP/R resolution (S2 Fig).

We attempted to perform survival analysis of the HVTN 910 data to assess the duration of VISP/R in this population. By 10 years of follow-up, 47.4% of participants enrolled in HVTN 910 saw VISP/R resolution (Fig 3A). When grouped by vaccine platform, the 10-year probability of VISP/R persistence was 63% (95% CI: 46–86%) for DNA vaccine recipients, 23% (95% CI: 18–29%) for VV vaccine recipients, 69% (95% CI: 64–76%) for DNA/VV vaccine recipients, and 97% (95% CI: 90–100%) for protein recipients (Fig 3B). Wide confidence intervals were typically a result of irregular participant visits and the uneven follow-up time.

Due to the association between *gag*, *gp140+* gene insert, and protein boost with VISP/R, we also assessed the duration of VISP/R in participants with or without these vaccine components. The 10-year VISP/R persistence rate was 6% (95% CI: 3–11%) for those receiving regimens

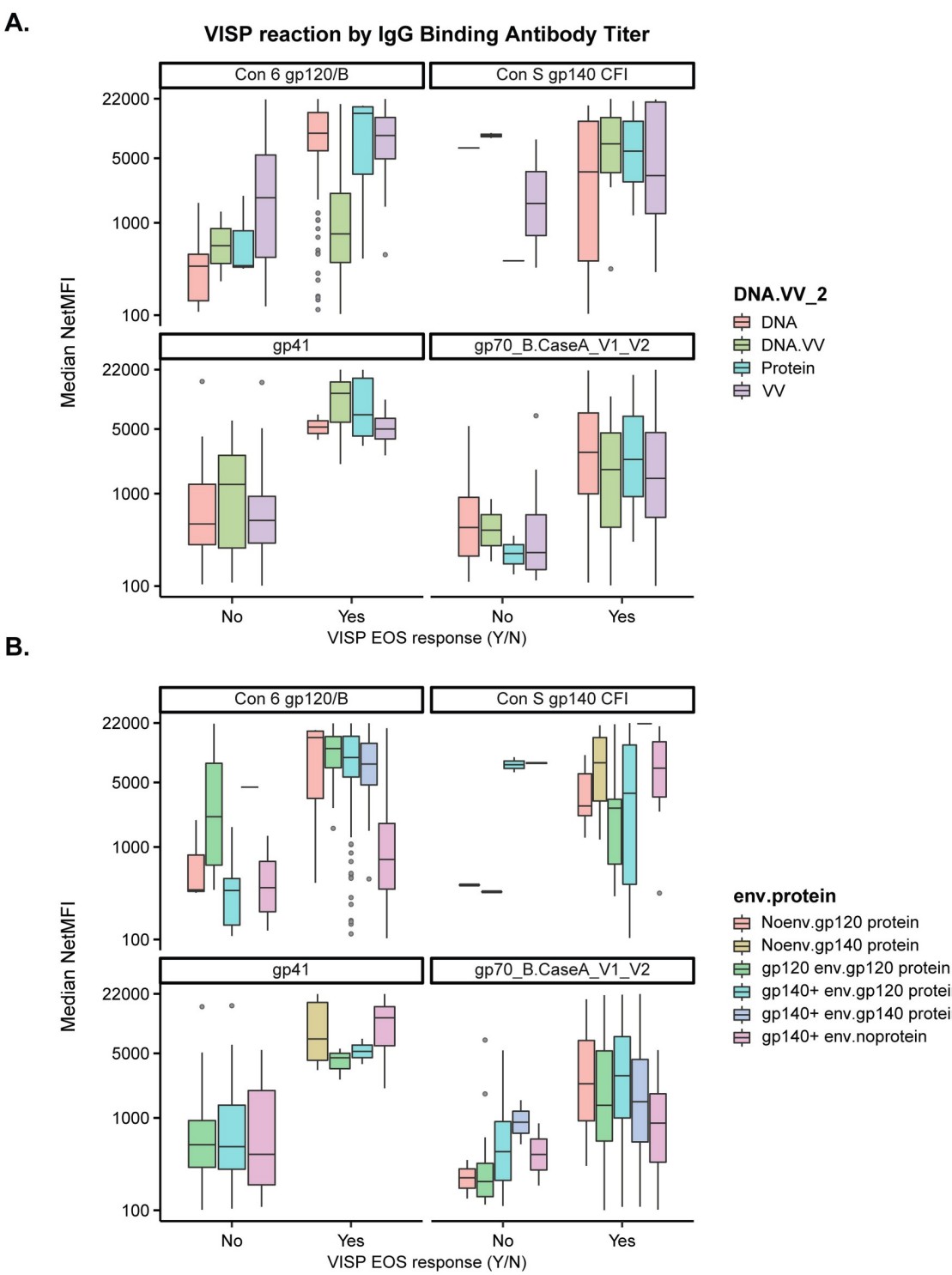

**Fig 2. IgG antibody responses of a subset of participants by VISP/R status.** Participants enrolled in HVTN 098, 100, 106, 107, 108, 111, 112, 114, 117, 118, 122, 205, 505 clinical trials have available IgG binding antibody magnitudes to antigens Con 6 gp120/B, Con S gp140 CFI, gp41, gp70_B.CaseA_V1_V2. Median IgG binding antibody (netMFI) was assessed at the peak time point, two to four weeks post last scheduled vaccination, by (A) product type and (B) envelope gene/protein combination. Gray circles represent individual BAMA netMFI results greater than 1.5x the interquartile range away from the median.

**Table 5. HVTN 910 study metrics.**

| | N* | Combined N = 1093 | Sub-Saharan Africa N = 230 | Americas/Europe/WP N = 863 |
|---|---|---|---|---|
| Years follow-up since HVTN 910 enrollment (Median) [min, 25%, 75%, max] | 1085 | 2.0 [0.0, 1.0, 4.0, 8.4] | 1.9 [0.0, 1.0, 3.2, 7.7] | 2.0 [0.0, 0.9, 4.2, 8.4] |
| 0** | | 11% (120/1085) | 13% (29/230) | 11% (91/855) |
| >0, <1 | | 15% (161/1085) | 10% (22/230) | 16% (139/855) |
| ≥1, <2 | | 25% (273/1085) | 30% (68/230) | 24% (205/855) |
| ≥2, <4 | | 24% (259/1085) | 31% (71/230) | 22% (188/855) |
| ≥4, ≤10 | | 25% (272/1085) | 17% (40/230) | 27% (232/855) |
| Years since parent protocol VISP/R assessment (Median) [min, 25%, 75%, max] | 1084 | 4.3 [0.0, 2.4, 7.0, 15.2] | 3.6 [0.0, 2.1, 5.8, 12.9] | 4.6 [0.1, 2.5, 7.1, 5.2] |
| 0 | | 1% (13/1084) | 6% (13/230) | 0% (0/854) |
| >0, <1 | | 3% (36/1084) | 5% (11/230) | 3% (25/854) |
| ≥1, <2 | | 14% (147/1084) | 13% (31/230) | 14% (116/854) |
| ≥2, <4 | | 27% (298/1084) | 33% (76/230) | 26% (222/854) |
| ≥4, <10 | | 50% (537/1084) | 40% (92/230) | 52% (445/854) |
| ≥10, <30 | | 5% (53/1084) | 3% (7/230) | 5% (46/854) |
| Years since last vaccination (Median) [min, 25%, 75%, max] | 1093 | 6.6 [0.8, 3.5, 8.3, 28.9] | 5.6 [1.0, 3.0, 7.7, 13.4] | 6.7 [0.8, 3.8, 8.5, 28.9] |
| >0, <1 | | 0% (2/1093) | 0% (0/230) | 0% (2/863) |
| ≥1, <2 | | 7% (78/1093) | 11% (26/230) | 6% (52/863) |
| ≥2, <4 | | 22% (239/1093) | 26% (60/230) | 21% (179/863) |
| ≥4, <10 | | 62% (679/1093) | 59% (135/230) | 63% (544/863) |
| ≥10, <30 | | 9% (95/1093) | 4% (9/230) | 10% (86/863) |
| Terminated from HVTN 910 | 1093 | 100% (1091/1093) | 100% (230/230) | 100% (861/863) |
| Death | | 0% (5/1093) | 0% (1/230) | 0% (4/863) |
| Early Termination | | 2% (19/1093) | 1% (2/230) | 2% (17/863) |
| HIV infection | | 2% (19/1093) | 4% (10/230) | 1% (9/863) |
| Relocated | | 5% (51/1093) | 2% (4/230) | 5% (47/863) |
| Resolution of VISP/R | | 17% (189/1093) | 45% (104/230) | 10% (85/863) |
| Site Closure | | 2% (19/1093) | 0% (0/230) | 2% (19/863) |
| Study Closure | | 67% (735/1093) | 40% (92/230) | 75% (643/863) |
| Unable to contact participant | | 5% (56/1093) | 7% (17/230) | 5% (39/863) |
| Median number of VISP/R assessments [25%, 75%] | 1093 | 5 [3, 8] | 5 [3, 6] | 5 [3, 9] |
| Median number of VISP/R reactive [25%, 75%] | 1093 | 4 [2, 7] | 2 [1, 4] | 4 [2, 8] |
| One negative test | 1093 | 25% (276/1093) | 63% (146/230) | 15% (130/863) |
| Two consecutive negative tests | 1093 | 21% (225/1093) | 52% (120/230) | 12% (105/863) |
| Three consecutive negative tests | 1093 | 18% (195/1093) | 46% (106/230) | 10% (89/863) |
| Three consecutive negative tests over 1 year | 1093 | 12% (132/1093) | 33% (76/230) | 6% (56/863) |

*Some parameters have totals <1093 resulting from missing data (e.g., evaluation of VISP/R date)

**0 years follow-up indicates that there was only one VISP/R-related visit, and no follow-up

containing no *env* and 65% (95% CI: 60–70%) for those receiving *gp140+ env* (Fig 3C). Of the few *gp120 env* recipients enrolled in HVTN 910, the 9-year VISP/R persistence rate was 33% (95% CI: 11–100%). When grouped by protein boost, the 10-year probability of VISP/R persistence was 45% (95% CI: 41–50%) for those receiving no glycoprotein, 73% (95% CI: 60–88%) for gp120 protein recipients, and 70% (95% CI: 61–81%) in all others (Fig 3D). The 10-year VISP/R persistence rate was 48% (95% CI: 44–53%) for those receiving regimens containing *gag* and 43% (95% CI: 33–56%) for those who did not (Fig 3E). Since the durations of VISP/R for gp120 and gp140+ gene and protein recipients were comparable, we assessed the

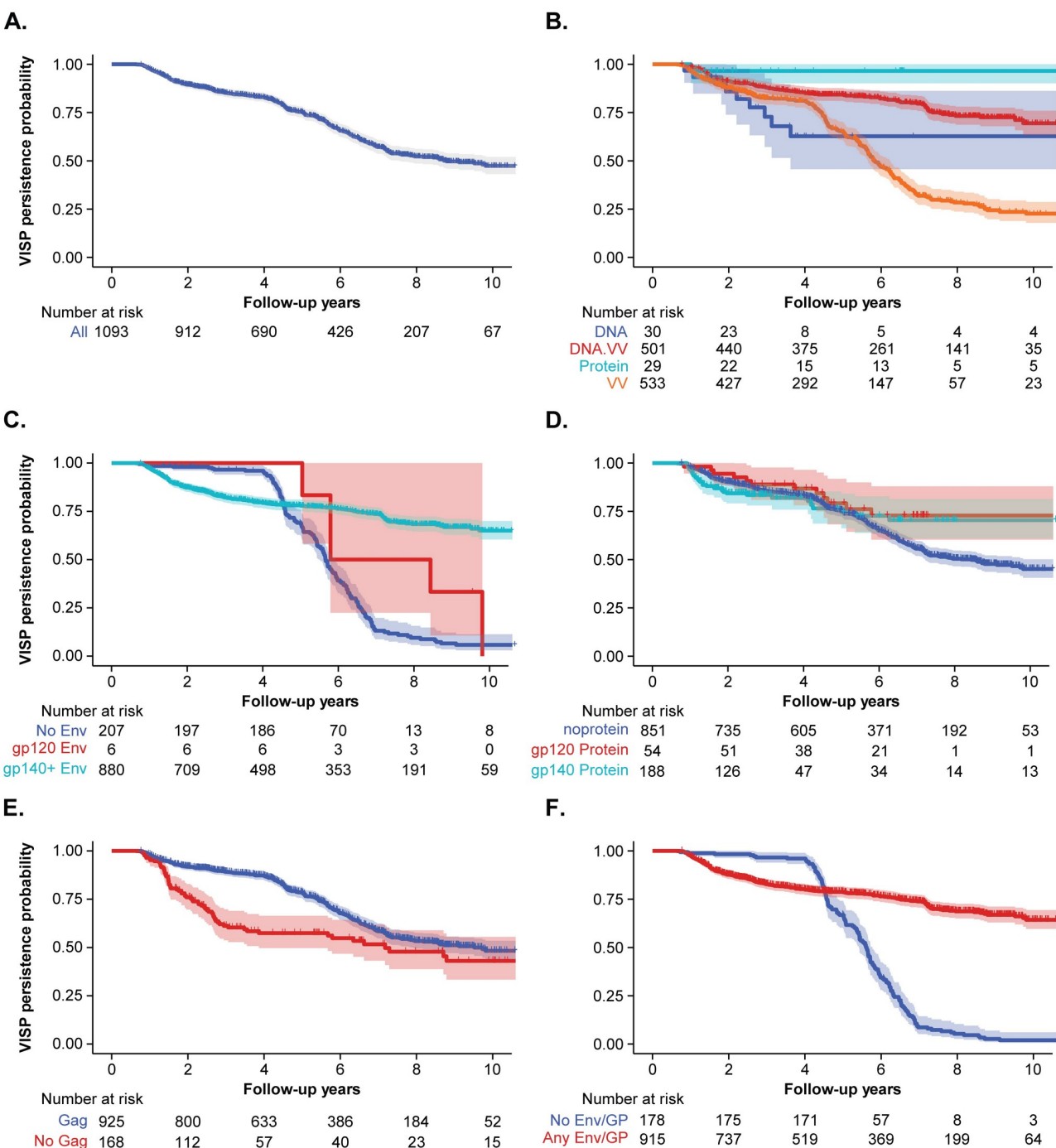

**Fig 3. HVTN 910 survival analysis by HIV vaccine regimen characteristic.** Kaplan-Meier (KM) estimates of HVTN 910 participants; shaded regions represent 95% CIs. KM estimates for (A) all HVTN 910 participants; and segregated by (B) vaccine platform, (C) *Env* gene, (D) Envelope protein, (E) *Gag* gene, and (F) Envelope gene or glycoprotein (GP). Participants were censored at time of their modified VISP/R resolution. "At risk" refers to number of participants observed at each timepoint that can contribute to the estimation of VISP/R persistence.

persistence of VISP/R for participants who did not receive envelope gene or glycoprotein (*gag* only) to those that received any envelope or glycoprotein (with or without *gag*) (Fig 3F). The 10-year VISP/R persistence rate was 2% (95% CI: 0.7–6%) for participants receiving no envelope or glycoprotein and was 64% (95% CI: 60–70%) for those that received any combination of envelope and/or glycoprotein.

### Assessing the VISP/R rate per diagnostic platform

Most serological HIV diagnostic tests are currently designed to detect Gag- and Env-specific antibodies. We measured the rate of VISP/R for each test used in the 75 HIV vaccine clinical trials in order to identify which assays most commonly resulted in the identification of VISP/R (Table 2). HIV diagnostic tests had variable rates of VISP/R in participants who only received *gag* gene inserts (1–70%). Most laboratory screening and rapid HIV tests had VISP/R rates >50% in participants who received gp140+ gene with or without gp140 protein (64–100%). None of these HIV tests had VISP/R rates >50% in participants who received gp140+ gene and gp120 protein (1–32%). The Abbott HIV AB HIV1/2 (rDNA) and bioMerieux Vironostika HIV-1 were the only laboratory screening or rapid tests that had VISP/R rates >50% in participants who received gp120 envelope or proteins. Although Western Blot/differentiation tests are not considered in the definition of VISP/R, 64–100% participants were reactive in these assays, regardless of the regimen received.

## Discussion

Researchers continue to evaluate and advance experimental preventative HIV vaccines that vary in platform, HIV antigens, and adjuvants. While primary analyses in this study evaluated 10-year persistence, it is noteworthy that we observed candidate HIV vaccines inducing VISP/R persistent in study participants nearly 30 years after trial participation. The probability of developing long-lasting VISP/R is mostly related to the vaccine regimen and HIV antigens included in the study product(s), with minimal impact from participant characteristics. This analysis showed that participants receiving viral vectors, viral vectors in combination with DNA vaccines, or proteins were more likely to have VISP/R at the end of the parent protocol than those receiving DNA vaccine regimens. In contrast, more participants who received DNA or DNA/viral vector combination vaccines still had VISP/R 10 years after vaccination than those only receiving viral vector vaccines. Although participant characteristics like geographic location and study phase were associated with increased odds of VISP/R in our analysis, these observations are likely due to operational decisions for study conduct, such as clinical research site selection and progression of immunogenic HIV vaccines from phase 1/2a to phase 2b studies.

The effect of gag in a vaccine regimen on inducing VISP/R was difficult to discern when administered in combination with regimens containing envelope or glycoprotein. It was noteworthy that a portion of participants who received regimens with only gag and no envelope or glycoprotein did have VISP/R at the end of their parent protocol, but only 2% of these participants had VISP/R 10 years after vaccination. Gag is therefore capable of inducing VISP/R, consistent with the presence of Gag antigens in commercial diagnostic tests, but Gag-induced VISP/R appears to be shorter-lived relative to envelope- or glycoprotein-induced VISP/R.

Participants receiving regimens with gp140+ gene inserts or proteins had higher rates of VISP/R at the end of their participation in a clinical trial, and higher persistence of VISP/R at 10 years than those who received any other product. Inclusion of gp120 envelope in a regimen increased the odds of VISP/R in recipients, but a gp120 protein boost, regardless of the type of *env* gene insert, reduced these odds. A hypothesis for this seemingly contradictory observation

could be that vaccine regimens with a gp140+-containing prime induces the presence of anti-envelope antibodies detected by common diagnostic tests, but a gp120 protein boost then diverts the immune response away from generating the cross-reactive antibodies that cause VISP/R.

Participants with VISP/R had high levels of antibody responses to gp120, gp140, V1V2 and gp41 antigens for all subsets of vaccine platforms. Uneven availability of BAMA data for every *env*/glycoprotein combination made inferences of antibody response by envelope gene insert of protein received difficult. However, results from Palli et. al. provide evidence that while participants receiving DNA and/or viral vector (ALVAC versus modified vaccinia Ankara) combinations with gp120 or gp140+ gene and protein components resulted in IgG and IgG3 antibody responses to these antigens, the gp140 and gp41 IgG responses had longer half-lives than gp120 responses [2]. These results suggest that the persistence of VISP/R in recipients of regimens containing gp140+ genes or proteins is related to the characteristics of the antibodies targeting gp140 and gp41 antigens, as generated by these vaccine regimens.

The induction and persistence of HIV antibodies is an unavoidable (and in fact, desirable) aspect of a successful vaccine candidate. The occurrence of VISP/R associated with these antibodies, however, is equally and fundamentally dependent on the design of commercial HIV tests in use. Many commercial HIV tests indicate HIV acquisition by detecting antibodies to the immunodominant epitopes in gp41 [11–13]. Reliance on this immunodominant domain improves the performance of these assays in detecting acute HIV acquisitions, but complicates the diagnoses of HIV status in former vaccine trial participants [6, 14, 15]. Since these antibody assays are cheaper and simpler to perform than assays that detect HIV nucleic acid (a more direct indication of HIV acquisition in a participant specimen), diagnostic laboratories continue to rely on detection of these antibodies despite the potential complications in assay interpretation. If HIV vaccine developers continue to include gp140 in candidate vaccine regimens, many more study participants may develop VISP/R and then require careful management of their diagnostic testing until a new paradigm in HIV testing is widely adopted in public health settings.

This study has several key limitations. Meta-analysis of these HIV vaccine trials resulted in a dataset that lacked VISP/R rates for all possible combinations of vaccine candidate characteristics. Variable enrollment numbers in vaccine trial arms also meant varied depth of data available for each vaccine characteristic combination, particularly for associations between VISP/R and antibody response. This, in addition to the uneven and incomplete participant follow-up, impacted the precision to which VISP/R persistence could be measured in HVTN 910. This analysis also did not identify the epitopes within gp140 that result in VISP/R, and could not differentiate how the conformation of the gp41 region in the gp140+ *env* or protein within the vaccine regimen affects immunogenicity. For example, if vaccinated study participants receive gp140+ *env* or protein in which the immunodominant domain of gp41 is exposed, we hypothesize that their immune system would direct antibodies to this region, and these antibodies would in turn result in VISP/R. As vaccine developers generate products that express stable glycoprotein trimers in which this same region is modified or occluded, the antigenicity of the immunodominant domain (and resulting likelihood of inducing VISP/R) may be lower, but this hypothesis remains untested. Lastly, although all presented analyses tried to adjust for confounding factors that could potentially influence the effect of the demographics and vaccine characteristics on the rate and durability of VISP/R, there could be other unmeasured factors that may contribute to the observed patterns in these data.

Despite these limitations, this study represents the most comprehensive analysis of VISP/R to date. Analysis of 75 HIV vaccine trials and long-term observation of VISP/R in HVTN 910 has indicated that HIV envelope antigens from gene inserts or protein boosts increase the rate of VISP/R in study populations. This effect is modulated in part by other vaccine

characteristics (e.g. vaccine platform, *gag* inserts). As long as HIV antibody-based diagnostic methods are in use, participants and study teams must be prepared to manage this VISP/R potentially for many years after product administration as long as HIV-antibody based diagnostic methods are in use. Vaccine and diagnostic test developers should use this analysis to identify strategies to mitigate this impact on study participants through long-term diagnostic testing support and the development of VISP/R-agnostic testing methods.

## Supporting information

**S1 Fig. Follow up time of HVTN 910 participants from last vaccination.** Participants enrolled in HVTN 910 were monitored for HIV infection during the active study period and monitored for HIV infection and VISP/R during the HVTN 910 active study period. Time of first VISP/R is assumed as date of last HIV vaccination, and participants were followed up until VISP/R resolution, loss to follow up, or study termination, whichever comes first. (TIF)

**S2 Fig. HVTN 910 Kaplan-Meier analysis of persistence of VISP/R.** Kaplan-Meier (KM) estimates of all HVTN 910 Participants; shaded regions represent 95% CIs. Participants were censored at time of their per-protocol VISP/R resolution. "At risk" refers to number of participants observed at each timepoint that can contribute to the estimation of VISP/R persistence. (TIF)

**S1 Table. Multivariate analysis of VISP/R rate.**
(DOCX)

**S2 Table. Binding antibody titers and VISP/R.**
(DOCX)

**S3 Table. HVTN 910 demographics.**
(DOCX)

## Acknowledgments

We would like to thank the participants and study staff participating in AVEG and HVTN trials. We would also like to thank the HIV diagnostic testing laboratories for performing the HIV diagnostic testing analyzed here. We would like to thank Janine Maenza and Mary Allen for their participation in the HVTN 910 study team and the ongoing follow-up of participants with VISP/R. We would also like to thank Gail Broder, Michelle Andrasik, Mindy Miner, Sam Robinson, and the HVTN Scientific Review Committee for helpful editing.

## Author Contributions

**Conceptualization:** Nicole Espy, Esther Kwara, Stephen R. Walsh, John Hural.

**Data curation:** Nicole Espy, Xue Han, Shannon Grant, Bharathi Lakshminarayanan, Jessica Andriesen.

**Formal analysis:** Nicole Espy, Xue Han, Shannon Grant, Esther Kwara, Kelly E. Seaton, Erin Goecker, Jessica Andriesen, Yunda Huang.

**Funding acquisition:** Julie McElrath, John Hural.

**Investigation:** Nicole Espy, Xue Han, Esther Kwara, Kelly E. Seaton, Georgia D. Tomaras, Erin Goecker, Stephen R. Walsh.

**Methodology:** Nicole Espy, Xue Han, Shannon Grant, Esther Kwara, Kelly E. Seaton, Georgia D. Tomaras, Jessica Andriesen, Yunda Huang, Stephen R. Walsh, John Hural.

**Project administration:** Nicole Espy, Michael Stirewalt, Georgia D. Tomaras, John Hural.

**Supervision:** Nicole Espy, Yunda Huang, Stephen R. Walsh, John Hural.

**Visualization:** Nicole Espy, Xue Han.

**Writing – original draft:** Nicole Espy, John Hural.

**Writing – review & editing:** Nicole Espy, Xue Han, Shannon Grant, Esther Kwara, Kelly E. Seaton, Georgia D. Tomaras, Erin Goecker, Julie McElrath, Jessica Andriesen, Yunda Huang, Stephen R. Walsh, John Hural.

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
