## [Decision Letter · Decision Letter 0]

16 Mar 2023

PGPH-D-22-02087

Cross-protocol assessment of induction and durability of VISP/R in HIV preventive vaccine trial participants

Dear Dr. Hural,

Thank you for submitting your manuscript to PLOS Global Public Health. After careful consideration, we feel that it has merit but does not fully meet PLOS Global Public Health’s publication criteria as it currently stands. Therefore, we invite you to submit a revised version of the manuscript that addresses the points raised during the review process.

The reviewers find that the manuscript is overall well written but needs minor revisions. For example, the reviewers suggest clarifying which vaccination arms/study products are being compared in your study, improving the clarity of the cohort description, and discussing the choice of antibody titer analyses in the Introduction section. The detailed comments are appended below.

We look forward to receiving your revised manuscript.

Kind regards,

Alex Schaefer, PhD

Associate Editor

Journal Requirements:

2. We do not publish any copyright or trademark symbols that usually accompany proprietary names, eg  ©, ®, ™  (e.g. next to drug or reagent names). Please remove all instances of trademark/copyright symbols throughout the text, including ® on page 24.

Additional Editor Comments (if provided):

Reviewers' comments:

Reviewer's Responses to Questions

**Comments to the Author**

1. Does this manuscript meet PLOS Global Public Health’s publication criteria? Is the manuscript technically sound, and do the data support the conclusions? The manuscript must describe methodologically and ethically rigorous research with conclusions that are appropriately drawn based on the data presented.

Reviewer #1: Yes

Reviewer #2: Yes

2. Has the statistical analysis been performed appropriately and rigorously?

Reviewer #1: Yes

Reviewer #2: Yes

3. Have the authors made all data underlying the findings in their manuscript fully available (please refer to the Data Availability Statement at the start of the manuscript PDF file)?

Reviewer #1: Yes

Reviewer #2: Yes

4. Is the manuscript presented in an intelligible fashion and written in standard English?

Reviewer #1: Yes

Reviewer #2: Yes

5. Review Comments to the Author

Reviewer #1: TITLE:

Cross-protocol assessment of induction and durability of VISP/R in HIV preventive

vaccine trial participants

AIM:

The study assessed the impact of HIV vaccine characteristics (e.g. vaccine platform, gag inserts, adjuvants etc) on the induction and persistence of VISP/R among HIV vaccine recipients. The authors also identified HIV vaccine components that are more likely to induce VISP/R and assessed the detection rate of VISR by antibody based diagnostic assays routinely used in diagnosis of HIV infection.

Minor comments

1. Line 36-39: It’s not clear which vaccination arms/ study products are being compared.

a) gp140+ env gene insert or protein Vs gp120 env or protein components?

b) gp140+ env gene insert or protein Vs participants who received no env?

c) Vs gp120 env or protein components Vs participants who received no env?

2. Lines 46-47: The sentence would read better if it started with “Nearly 40 years after the discovery of HIV, the virus continues to disproportionately burden vulnerable countries and groups ….…

3. Line 48…add two words;

a) that before 1.5 million

b) occurred before in 2020

4. Line 113: define CDC since it is the first time it appears in the manuscript.

5. To improve clarity, I would suggest moving the text between lines 118-127 to appear under the description of the cohort. Merge the information in that paragraph with the cohort description

6. Lines 133-137: Since a total of 8,155 participants were tested for VISP/R out of 21,578 participants initially included in the analysis, it would be good to know how many of the 21,578 were placebo recipients, how many acquired HIV infection during follow up and how many did not complete the studies. This would help to understand why only about a third of the participants were tested for VISP/R at the end of their parent protocols.

7. The text in lines 140-145 does not match with what is presented in Fig 2 (the flow diagram). Fig 2 shows that 4290 participants had evidence of VISR but of these only 1146 were enrolled into HVTN910. What happened to the 3144 participants (4290-1146)? Why weren’t they enrolled into HVTN910? Moreover, the 4290 are not mentioned anywhere in the cohort description, they just pop up in Fig 2.

8. Lines 148-152: The fig 2 legend does not match the information in fig 2. The AVEG, HIVNET and HVTN 032, 034, 050, 057, 077, 502, 503 participants are not shown in Fig 2.

9. Line 156: DAIDS and GCLP appear for the first time in the manuscript. They need to be defined.

10. Line 160: CTU appears for the first time in the manuscript. They need to be defined.

11. Line 168; FDA and CE appear for the first time in the manuscript. They need to be defined.

12. Table 3: The probability of developing VISP/R is not related to participant characteristics. It’s highly influenced by the vaccine characteristics and the immunization schedule. I wonder why participants’ results have been split into two groups; sub-Saharan Africa vs the rest of the World. It would be good if the authors had provided justification for splitting the analysis of the data into those two groups. Additionally, I would suggest the authors add a column for overall (combined) results in the table, so the reader sees an overall picture first before going into the subgroups.

13. Table 4: add “s” to the word odd in the column containing multivariate odd ratio

14. Line 262: Add “the” before risk of VISP/R by binding antibody titers to Env

15. Line 287: EOS appears for the first time. It should be defined.

16. Table 5:

a) Provide overall (combined) results first

b) Provide justification for splitting the results into two groups

c) There should be an explanation below the table explaining why for some parameters the total number of volunteers is less than 1093 (i.e 1085 and 1084)

d) HVTN910 should NOT be written as 910

e) Median years since 910 enrollment [IQR].. what does it stand for? follow-up time since enrollment in HVTN 910? It’s not clear

f) Median years since 910 enrollment (%)… same comment as above

g) What is the difference between 0 and 0-1? Why shouldn’t the authors present the data as <1 year?

h) The content would be clearer if both actual number and % were used in the table 5 i.e 4 (X%).

i) Years since EOS [IQR]… Same comments as in e above

j) Years since last vaccination [IQR]… Same comments as in e above

k) Median N of VISP assessment. what does N mean? Also add (IQR)

l) Three consecutive Ns over 1 yr… What does Ns and Yr mean?

m) Ensure uniformity in presentation of data. The numbers and percentages in the table should either be centred, aligned to the right or to the left.

17. Lines 308-334: Maintain consistency. Indicate 95% CI inside the parenthesis of all VISP/R results. i.e 63% (95% CI: 46-86%), 23% (18-29%), 69% (64-76%), 97% (90-100%).

18. Lines 353-355: it’s written that the study “found that some HIV vaccine candidates have induced VISP/R in study participants lasting between 10-30 years”. But the participants were only followed up for a maximum of 10 years only. The authors should rephrase the statement to reflect the results in their manuscript.

19. Line 407: an ‘a’ is missing in the word metanalysis.

Reviewer #2: The manuscript reports the long-term results of induced seropositivity in HIV vaccine recipients in 75 studies in several continents. This issue may not be known to the PLOS Global Public Health readership but is addressed here by the largest study ever conducted.

Overall, the results are quite complex but well presented. However, there are some points that the authors need to address.

Minor comments:

Figure 1: Sorry to say that it is not very informative. The right area of the graph with the rapid test image does not describe the VISP algorithm and is quite confusing.

Line 106: Do the authors mean that gp120 and gp41 were grouped together? Please specify as the groupings are not apparent in Table 1.

Line 119: Unscheduled testing, but when were they recommended ?

Line 179: Antibody titer data appear at this point in the manuscript. What are the objectives in this study? Please discuss in the Introduction and briefly justify the choice of specific IgG antibodies instead of just giving references.

Line 244-245: A single p-value does not allow to conclude that the 3 platforms are different from DNA platform. My suggestion is to add the confidence intervals of the multivariate odds ratios in Table 4.

Line 256: OR=1.634 in the text but OR=0.752 in SI Table 1. Please clarify.

Figure 3: Gray circles are outside 1.5 x interquartile range. Please check the legend.

Table 5: For times since vaccination and since EOS, the 4+ year class is not very informative. We would have seen more extreme classes and the range in addition to the IQR.

Line 354: ‘10-30 years of VISP/R’, this is not shown in the results (see Table 5 comment)

Line 369: ‘98% no longer had VISP/R 10 years after” , not in the Results , except in Figure S1 but only graphically

Figure S2: Individual points are beyond the “outliers” in Figure 3. Please check or justify.

6. PLOS authors have the option to publish the peer review history of their article (what does this mean?). If published, this will include your full peer review and any attached files.

**Do you want your identity to be public for this peer review?** For information about this choice, including consent withdrawal, please see our Privacy Policy.

Reviewer #1: No

Reviewer #2: No

---

## [Decision Letter · Decision Letter 1]

17 May 2023

Cross-protocol assessment of induction and durability of VISP/R in HIV preventive vaccine trial participants

PGPH-D-22-02087R1

Dear Dr. Hural,

We are pleased to inform you that your manuscript 'Cross-protocol assessment of induction and durability of VISP/R in HIV preventive vaccine trial participants' has been provisionally accepted for publication in PLOS Global Public Health.

Best regards,

Julia Robinson

Executive Editor

Reviewer Comments (if any, and for reference):

Reviewer's Responses to Questions

**Comments to the Author**

1. If the authors have adequately addressed your comments raised in a previous round of review and you feel that this manuscript is now acceptable for publication, you may indicate that here to bypass the “Comments to the Author” section, enter your conflict of interest statement in the “Confidential to Editor” section, and submit your "Accept" recommendation.

Reviewer #1: All comments have been addressed

Reviewer #2: All comments have been addressed

2. Does this manuscript meet PLOS Global Public Health’s publication criteria? Is the manuscript technically sound, and do the data support the conclusions? The manuscript must describe methodologically and ethically rigorous research with conclusions that are appropriately drawn based on the data presented.

Reviewer #1: Yes

Reviewer #2: Yes

3. Has the statistical analysis been performed appropriately and rigorously?

Reviewer #1: Yes

Reviewer #2: Yes

4. Have the authors made all data underlying the findings in their manuscript fully available (please refer to the Data Availability Statement at the start of the manuscript PDF file)?

Reviewer #1: Yes

Reviewer #2: Yes

5. Is the manuscript presented in an intelligible fashion and written in standard English?

Reviewer #1: Yes

Reviewer #2: Yes

6. Review Comments to the Author

Reviewer #1: No further comments

Reviewer #2: (No Response)

7. PLOS authors have the option to publish the peer review history of their article (what does this mean?). If published, this will include your full peer review and any attached files.

**Do you want your identity to be public for this peer review?** For information about this choice, including consent withdrawal, please see our Privacy Policy.

Reviewer #1: No

Reviewer #2: No
